# 1 Characterization of a High Detection-Sensitivity Atmospheric

# Pressure Interface Time-of-Flight Mass Spectrometer

- Fabian Schmidt-Ott<sup>1,2</sup>, Somnath Bhowmick<sup>1</sup>, Alexandros Lekkas,<sup>3</sup> Dimitris Papanastasiou<sup>3</sup>,
- Anne Maisser<sup>1</sup>, George Biskos<sup>1,4</sup>
- <sup>1</sup> Climate and Atmosphere Research Centre, The Cyprus Institute, 2121 Nicosia, Cyprus
- <sup>2</sup> Institute for Atmospheric and Earth System Research, University of Helsinki, 00014, Helsinki, Finland
- <sup>3</sup> Fasmatech Science and Technology SA, TESPA Lefkippos, NCSR Demokritos, 15310 Athens, Greece
  - <sup>4</sup> Faculty of Civil Engineering and Geosciences, Delft University of Technology, 2628 CN, Delft, The
- Netherlands

Correspondence to: Fabian Schmidt-Ott (f.schmidt-ott@cyi.ac.cy) or George Biskos (g.biskos@cyi.ac.cy or g.biskos@tudelft.nl)

**Abstract.** We have characterised a new Atmospheric-Pressure-interface Time-of-Flight Mass Spectrometer, equipped with an octapole ion trap for accumulating the sampled ions before orthogonally accelerating them into the mass analyser. The characterisation has been carried out using ion standards produced by electrospray ionisation, that were subsequently mobility-selected by a differential mobility analyser operated at atmospheric pressure. Our results show that the detection sensitivity (or limit of detection) of the mass spectrometer is in the parts per quintillion (i.e.,  $10^{-3}$  parts per quadrillion, ppq; which is  $\sim 30$  ions/cm<sup>3</sup>) range with temporal resolutions of 1 second. When increasing the temporal resolution up to 1 minute, the detection sensitivity can be reduced to the 10 parts per sextillion (i.e.,  $10^{-5}$  ppq; which is  $\sim 0.3$  ions/cm<sup>3</sup>) range, enabling the system to measure gaseous ions of extremely low concentrations. In contrast to other mass spectrometers that employ spectra accumulation to improve the detection sensitivity for atmospheric observations, ion accumulation amplifies the signal without increasing the noise level; something that among others is highly important for probing short-lived ionic clusters during new particle formation events in the atmospheric environment. We also show that the mass spectrometer has a transmission of up to 1%, and a mass resolution of 23,000 for ionic masses of ca. 600 Da., while it can be used in ways to induce collision dissociation of the sampled ions by tuning the operating conditions of the Atmospheric-Pressure-interface stage.

# 1. Introduction

Time-of-Flight Mass Spectrometry (TOF-MS) has proven to be an effective method for chemical identification of species, both in field and in laboratory environments. Since its introduction in the 1940s (Stephens 1946), TOF-MS has been employed in combination with different ionization techniques and sampling interfaces, including, among others, Extractive Electrospray Ionization (EESI; Chen et al., 2006), Proton Transfer Reaction (PTR; Hansel et al., 1995), and Matrix-Assisted Laser Desorption/Ionization (MALDI; Karas et al., 1987), or with Atmospheric Pressure interface (APi). The latter has emerged as a powerful tool for measuring ionic species in gases, including in the ambient air where it has proved to be a valuable system for understanding nucleation in the atmospheric environment. APi-TOF-MS systems that have been described in the literature are manufactured by Aerodyne Research Inc. (Billerica, Massachusetts, USA) and Tofwerk AG (Thun, Switzerland), having resolutions > 2000, m/z ranges up to 10,000, and transmissions from 0.1 to 1% depending on the mass-to-charge ratio of the sampled ions (Junninen et al., 2010; Leiminger et al., 2019).

A key challenge in using APi-TOF-MS systems in field studies of atmospheric nucleation is that their detection sensitivity (or limit of detection; LOD), which is proportional to the Signal-to-Noise Ratio (SNR), has to be high enough to detect trace concentrations of molecules that potentially nucleate to form clusters, and subsequently nanoparticles upon condensation and/or coagulation, if conditions are favourable (Kulmala et al., 2004). For

instance, measuring precursor gases that play a key role in atmospheric nucleation, such as  $H_2SO_4$ , requires APi-TOF-MS systems that have an LOD in the parts-per-quadrillion (ppq) range (Beck et al., 2022), whereas it has to be even below  $10^{-3}$  ppq (~ 30 ions/cm<sup>3</sup>) for probing the evolution of the initial clusters (Kulmala et al., 2013).

Another reason for improving the detection sensitivity of MS systems is to enable coupling with other instruments for elaborating the measurements, such as a Differential Mobility Analyzer (DMA; Hogan & de la Mora, 2011) that filters out a big fraction of the sampled ions based on their mobility, and/or ionisation sources that have low efficiency. For example, the ionization efficiency of  $C_6H_8O_5$  by nitrate-based chemical ionization (CI) is approximately  $10^{-7}$  (Hyttinen et al., 2015). Combined with the inherently low concentrations of atmospheric trace gases, CI systems have to be coupled with mass spectrometers that have high detection sensitivity (i.e., low LOD) to detect and probe the evolution of  $C_6H_8O_5$  (or other species of similar importance) in the atmospheric environment during nucleation events.

A common practice for lowering the LOD in APi-TOF-MS systems is to accumulate multiple mass spectra over extended sampling periods. In this case, the ion signal grows linearly with the number of accumulated spectra (n), while the noise increases proportionally with the square root of n. Using this approach, Kürten et al. (2011) have shown that the LOD of their CI-APi-TOF-MS system (i.e., a system that includes a chemical ionization stage) is between 0.4 and 2.4 ppq (which is in the range of  $10^{-2}$  molecules/cm<sup>-3</sup>) with a sampling time ( $T_s$ ) of 1 minute for the measurement of neutral  $H_2SO_4$ . When  $T_s$  is increased to 1 hour, Junninen et al. (2010) have shown that the LOD of their APi-TOF-MS system, produced by Tofwerk AG, is  $2.4 \times 10^{-5}$  ppq ( $\sim 0.7$  molecules/cm<sup>-3</sup>). Using a similar approach, Leiminger et al. (2019) reported that the LOD of the ioniAPi-TOF MS (Ionicon Analytik GmbH) is  $2 \times 10^{-8}$  ppq ( $6 \times 10^{-4}$  molecules/cm<sup>-3</sup>) with the same  $T_s$ . Although such LOD values are extremely low and can thus enable detection of precursor gases in ambient air, the sampling times are too long to probe the evolution of fast-evolving species during nucleation events.

An alternative approach for lowering the LOD, or enhancing the SNR, is to physically accumulate the ions within an ion trap placed upstream of the TOF chamber, yielding a signal amplification without increasing the noise level. In fact, the SNR is linearly proportional to the accumulation time ( $T_a$ ), assuming that the losses in the ion trap are negligible. This is a plausible assumption, as previous works have shown that extremely high trapping efficiencies in multipoles are possible (Pedersen et al., 2002; Xu et al., 2024). To our knowledge, the use of ion traps in APi-TOF-MS for improving their LOD has not yet been demonstrated.

Here, we characterize a novel APi-TOF-MS, manufactured by Fasmatech, Athens, Greece, that includes an octapole ion trap, and investigate how ion accumulation within the octapole influences the detection sensitivity of the instrument. In addition, we characterize its transmission efficiency, mass resolution, and ability for controlled Collision-Induced Dissociation (CID) of the sampled species.

# 2. Methods

2.1 The APi-TOF-MS

# The APi-TOF-MS system, the layout of which is shown in Fig. 1, has been previously described by Kaltsonoudis et al. (2023). Ions are introduced into the mass spectrometer through a capillary inlet (internal diameter of 0.5 mm) at a constant flow rate of 1.7 lpm, and guided through a series of Radio-Frequency (RF) ion guides to an orthogonal accelerator that periodically pulses them into the TOF chamber. The vacuum within the APi comprises four consecutive stages, progressively lowering the pressure by nine orders of magnitude from atmospheric levels

four consecutive stages, progressively lowering the pressure by nine orders of magnitude from atmospheric levels to 10<sup>-6</sup> mbar within the TOF chamber. More specifically, the APi-TOF-MS consists of an aerolens and an ion funnel, an RF octapole ion trap, an RF hexapole ion guide, a high-vacuum lens, and the TOF mass analyser.

The aerolens and the ion funnel are at the first vacuum stage (1 mbar) for laminarising the under-expanded gas flow and focusing the ions through a 2-mm differential aperture, respectively, downstream the capillary inlet. Ions are confined radially by fields created by a pair of antiphase RF signals applied to the aerolens and ion funnel

electrodes, with a frequency of 2.43 MHz and an amplitude of 70 V. An axial electric field is established by applying a direct current (DC) potential difference between the two elements, while the DC potential applied to the aperture is switched between transmission and deflection levels for gating ions further downstream the system (Papanastasiou et al., 2021).

Ions focused and gated through the differential aperture at the exit of the ion funnel are captured radially in an RF octapole ion guide operated at a pressure of  $10^{-3}$  mbar, with user-defined RF frequency and amplitude  $V_{RF,2}$  as specified in Table 1. The ion guide further confines and effectively brings the ion beam to the next vacuum stage of the MS. Lower RF frequencies and higher amplitudes enhance the transmission of higher m/z ions. The ion guide is segmented axially, and a weak DC gradient allows for efficient thermalization and axial transfer of the ions to downstream ion optics. The DC potential applied to the differential aperture installed at the beginning (Gate 1) and the end (Gate 2) of the RF octapole, as shown in Fig. 1, can be adjusted, enabling the accumulation and subsequent release of the sampled ions towards the TOF mass analyser. The dual gate configuration allows both for adjusting the ion load inside the RF octapole ion guide, as well as controlling the release of ion packets towards the TOF mass analyser (see Fig. 1).

Ions are radially confined by an RF-only hexapole ion guide operated at  $10^{-5}$  mbar, followed by a two-stage Einzel lens configuration coupled to a dual 2-mm slit for shaping the ion beam entering the TOF analyser. The dual Einzel lens is designed to transform the cylindrical cross-section ion beam at the exit of RF hexapole into a two-dimensional beam so that they can be accelerated by the Orthogonal Acceleration (OA) region into the TOF (Kaltsonoudis et al., 2023).

The last stage of the system is the TOF mass analyser that comprises a two-stage OA region and a two-stage reflectron operated at a voltage of ~9 kV. The high-voltage extraction pulses applied across the first stage of the OA are synchronized with the release of the ions from the RF octapole ion guide. We should note here that heavier ions arrive into the effective region of the OA later than the lighter, causing a mass-dependent axial spread of the ions. To account for that, multiple OA pulses, with different delays relative to the ion guide release, are used to efficiently sample each packet having ions of a broad m/z range. Precise adjustment of the delay between ion release and OA pulsing is therefore required to ensure efficient sampling across the entire mass range while minimizing mass discrimination (see Supplementary Information for details).

Ions are time-focused on a MagneTOF Electron Multiplier Detector (ETP; Sydney, Australia) operated at a gain of 1900 V. The length of the TOF chamber is ca. 2 m and the pressure is maintained at  $10^{-6} \text{ mbar}$ .

Figure 1: Schematic layout of the experimental setup, including all stages of the APi-TOF-MS system. Ions were generated by ESI and mobility-selected by the pp-DMA. The resulting monodisperse ions were then directed to both a FCE and the APi-TOF-MS system. The graph at the bottom shows the axial voltages applied across various stages of the APi-TOF-MS system, where gates 1 and 2 are the entrance and exit of the ion trap, respectively.

Table 1. Optimised operating configuration of the APi-TOF-MS, for high- and low-mass ranges.

| Stage      | Input           | High-mass       | Low-mass       |
|------------|-----------------|-----------------|----------------|
| Ion Eumal  | RF              | 2.36 MHz        | 2.36 MHz       |
| Ion Funnel | $V_{RF,1}$ 70 V |                 | 70 V           |
| Octapole & | RF              | 2.36 MHz        | 2.36 MHz       |
| Hexapole   | $V_{RF,2}$      | 500 V           | 250 V          |
| TOF        | OA delay        | 60, 140, 220 μs | 60, 90, 120 μs |
| chamber    | MagneTOF gain   | 1900 V          | 1900 V         |

#### 2.2 Experimental Setup and Procedure

Fig. 1 shows the experimental setup employed for all the measurements carried out in this work. Ions of well-defined mobilities and masses were produced by electrospray ionization (ESI) from acetonitrile-based solutions containing ~10 mmol/L of tetraalkyl ammonium halide salts (see Table 2; Ude & De La Mora, 2005). The ESI was coupled to a parallel plate DMA (pp-DMA; SEADM P5 DMA), the front plate of which was electrically floated, in order to classify either the monomer or dimer of the halide salt. The ESI was operated in a counterflow mode, where a gas flow of 1.2 lpm coming from the monodisperse inlet of the ppDMA opposed the electrosprayed solution flow, in order to remove droplets and neutral species as well as to maximize the resolution of the DMA (Amo-González & Pérez, 2018). The resolving power of the pp-DMA, determined as the full width half maximum (FWHM) divided by the peak mobility, was around 50 for the THAB monomer.

The concentration of ions downstream the pp-DMA was measured using a Faraday Cup Electrometer (FCE; Lynx E12, SEADM), placed in parallel to the APi-TOF-MS system. The length of the tubing from the DMA to either the APi-TOF-MS system or the FCE and the flow rates through them were the same (10 cm and 1.7 lpm, respectively), ensuring equal ion losses upstream the two instruments. In all measurements, ion production was kept as stable as possible, with a maximum relative standard deviation of the ion concentration within 10% for different ion production rates (see Fig. S-2 in the supplement). When not desired, fragmentation within the APi-TOF-MS system was minimized by adjusting the potential between the ion funnel and the ion trap. In this way fragmentation was reduced at levels where the signal corresponding to fragmented species was down to 3% of the signal corresponding to the original compound.

Table 2. Positively and negatively charged ions produced by ESI. Key: TMAI: tetra-methyl ammonium iodine; TPAI: tetra-propyl ammonium iodide; TBAI: tetra-butyl ammonium iodide; THAB: tetra-heptyl ammonium bromide; TDAB: tetra-decyl ammonium bromide; TDDAB:

tetra-dodecyl ammonium bromide.

| Species                |                         | m (Da)  | Z (cm <sup>2</sup> /Vs) |
|------------------------|-------------------------|---------|-------------------------|
| TMAI monomer           | $C_4H_{12}N^+$          | 74.10   | 2.18                    |
| TMAI dimer             | $(C_4H_{12}N)_2I^+$     | 275.10  | 1.48                    |
| TPAI monomer           | $C_{12}H_{28}N^{+}$     | 186.22  | 1.62                    |
| TPAI dimer             | $(C_{12}H_{28}N)_2I^+$  | 499.35  | 0.99                    |
| TBAI monomer           | $C_{16}H_{36}N^{+}$     | 242.29  | 1.39                    |
| TBAI dimer             | $(C_{16}H_{36}N)_2I^+$  | 611.47  | 0.87                    |
| THAB monomer           | $C_{28}H_{60}N^{+}$     | 410.47  | 0.97                    |
| THAB dimer             | $(C_{28}H_{60}N)_2Br^+$ | 899.86  | 0.65                    |
| TDAB monomer           | $C_{40}H_{84}N^{+}$     | 578.66  | 0.78                    |
| TDAB dimer             | $(C_{40}H_{84}N)_2Br^+$ | 1236.24 | 0.54                    |
| TDDAB monomer          | $C_{48}H_{100}N^{+}$    | 690.79  | 0.71                    |
| Bromide ion            | Br -                    | 78.92   | N.A.                    |
| Iodide ion             | I-                      | 126.90  | N.A.                    |
| Potassium diiodide ion | $I_2K^-$                | 292.77  | N.A.                    |

The LOD of the APi-TOF-MS system is equivalent to the inlet concentration that can be readily distinguished from noise level, which in our case corresponds to the signal of a single ion striking the TOF detector ( $S_{ion} = 60$  a.u.; see Fig. S-3). In this study,  $T_a$  was varied from 0.001 to 50 s for different inlet ion concentrations ranging from 10 down to  $10^{-5}$  ppq ( $3 \times 10^5 - 0.3 \text{ ions/cm}^3$ ). Such low values were produced both by lowering the concentration of the solvent in the ESI solution down to the limit at which the electrospray was still stable, and by increasing the distance between the tip of the ESI capillary and the DMA inlet ( $d_{cap}$ ; see Fig. 1), effectively lowering the probability of ions passing through the DMA inlet slit. The lowest ion concentration that could be produced by ESI was therefore limited by the distance  $d_{cap}$  (see Fig. 1), which could be set up to 25 mm.

To determine the mass-dependent transmission of the APi-TOF-MS, ions listed in Table 2 were generated at concentrations of approximately 0.1 ppq ( $3 \times 10^3$  ions/cm<sup>3</sup>), while operating the APi-TOF-MS system at either the high-mass or low-mass setting (see Table 1), and a  $T_a$  in the ion trap of 0.01 s. Low ion concentrations and short accumulation times are critical for this measurement in order to minimize space-charge effects that affect transmission through the ion trap or possibly other stages within the APi-TOF-MS system. The transmission of the instrument is defined as the ratio between the ion count rate, expressed in ions/s, measured by the APi-TOF-MS system ( $I_{MS}$ ), to that measured by the electrometer ( $I_{FCE}$ ), i.e.:

$$T = \frac{I_{MS}}{I_{FCE}},\tag{1}$$

where  $I_{MS} = \frac{S}{n \cdot S_{ion} \cdot T_a}$ , with S being the signal recorded by the APi-TOF-MS system and n the number of accumulated spectra, set to 10 in all our measurements.

The mass-dependent resolution of the system was determined by a mass spectrum corresponding to sulphuric acid-nitrate clusters, produced by electrospraying a methanol–ammonium sulphate solution, following the procedure described by Waller et al. (2019), as shown in Fig. S-4. The mass resolution of the API-TOF-MS system at a specific m/z value is defined as:

$$R = \frac{m/z}{FWHM} \,. \tag{2}$$

In a final series of measurements, we investigated the CID of THAB and TPAI dimers occurring between the ion funnel and the ion trap where the pressure is reduced from 1 to  $10^{-3}$  mbar, offering optimal conditions for ion fragmentation (Zapadinsky et al., 2019). We should note here that CID is used to probe the declustering strength of the sampled compounds; e.g., for assessing the compatibility of reagent ions with analyte molecules in chemical ionization (Brophy & Farmer, 2016; Lopez-Hilfiker et al., 2016). Key parameters influencing the dissociation of molecules or clusters include the pressure within the collision cell, the type of neutral gas molecules, the kinetic energy of the ions as well as their nature and bonding energies among their atoms (Hyttinen et al., 2018; Sleno & Volmer, 2004). In our measurements, dissociation was controlled by adjusting the potential between the ion funnel and the ion trap (see Fig. 1), thereby increasing or decreasing the kinetic energies of the ions over or below their dissociation threshold points.

#### 3. Results and Discussion

The following paragraphs provide our results and a discussion on the detection sensitivity (section 3.1), the transmission (section 3.2), the mass resolution (section 3.3) of our APi-TOF-MS system, as well as the collision-induced dissociation (section 3.4) of clusters sampled by it.

#### 3.1 Detection Sensitivity

Figure 2 shows the signal of the APi-TOF-MS system recorded when feeding it with THAB monomers at different concentrations, and trapping them in the ion trap for specific accumulation times  $(T_a)$ . As mentioned above, all these measurements are averages of 10 spectra in order to reduce uncertainties from fluctuations in ion production from the electrospray source. The signal increases linearly with the ion concentration up to approximately  $S=10^6$  a.u., indicating concentration-independent transmission up to this limit for a given ionic mass. Beyond this limit, the signal plateaus for all  $T_a$  values, suggesting that space-charge effects become significant within the ion trap, leading to a maximum concentration of ions that can be accumulated (Majima et al., 2012). The space-charge limit is therefore reached either when introducing too high ion concentrations or when using an accumulation time that is too long.

 The similar slopes of all curves in Fig. 2 indicate that ion transmission remains nearly constant across the range of  $T_a$  values tested, suggesting that the ion trap is highly effective provided that the saturation limit is not reached. These findings are consistent with previous studies reporting that losses in multipole ion traps can be extremely low (Pedersen et al., 2002; Xu et al., 2024).

Figure 2: Curves of the APi-TOF-MS system signal as a function of ion concentration at the inlet of the instrument, for specific  $T_a$  values within the ion trap. Solid symbols represent the measurements in which the signals from both the FCE and the APi-TOF-MS system are within detection range, while the open symbols indicate measurements in which only signals from the APi-TOF-MS system are so. For the measurements reported here we used THAB monomer ions. Arrows indicate ion concentrations corresponding to the LOD for each  $T_a$  used. The fitted curves for S <  $10^6$  are of the form  $\log(S) = a \cdot \log(N_{lon}) + b \cdot \log(T_a) + c$ , where a = 1.56, b = 1.30 and c = 7.60

The limit of detection of the Faraday cup electrometer (LOD $_{FCE}$ ) used in our setup to measure inlet concentrations was approximately  $3 \times 10^{-2}$  ppq (800 ions/cm $^3$ ). It was therefore not possible to directly investigate the LOD of the APi-TOF-MS below this concentration. However, it was possible to extrapolate inlet ion concentrations even below the LOD $_{FCE}$ , assuming that the transmission of the APi-TOF-MS is independent of concentration, and thus that the relationship between the ion concentration and the measured signal strength remains linear. The assumption that the transmission of the APi-TOF-MS remains constant even below the LOD $_{FCE}$  is reasonable considering that losses through the system in this low-concentration range are concentration-independent.

The limit of production of the electrospray ionization source (LOP<sub>ESI</sub>) was reached at  $2 \times 10^{-3}$  ppq (50 ions/cm<sup>3</sup>). At that point, the APi-TOF-MS remained well within its detection range even when using an accumulation time of only 0.1 s. This demonstrates the capability of the system to measure extremely low concentrations by further increasing  $T_a$ , and demonstrates that even lower concentrations than that achieved at the LOP<sub>ESI</sub> can produce a signal at the APi-TOF-MS system when increasing the accumulation time in the ion trap to a few minutes.

Figure 3 shows how the LOD of the APi-TOF-MS system decreases with increasing measurement time (or temporal resolution of the instrument) defined as  $\Delta t = n \cdot T_a$ . Evidently, by setting the temporal resolution to 500 s (i.e., 8.3 minutes) the LOD was reduced to  $4 \times 10^{-6}$  ppg (~0.1 ions/cm<sup>3</sup>).

For measurements in the atmospheric environment, concentrations of different ionic species can vary by several orders of magnitudes. At higher concentrations (i.e., in the  $10^4$  ions/cm³ range), it would be ideal to include such filtering system upstream of the APi-TOF-MS to prevent additional space-charge build-up induced by species that are not of interest within the ion trap. At lower concentrations (i.e., in the  $10^2$  ions/cm³ range), however, using a DMA as a filter for mass spectrometry should be considered with caution due to losses this can cause.

10<sup>-1</sup> 10<sup>3</sup> 10<sup>-2</sup> 10<sup>2</sup> 10<sup>-3</sup> LOD (ppq) 10<sup>0</sup> 10<sup>-5</sup> 10<sup>-1</sup> 10<sup>-6</sup> 10<sup>-2</sup> 10<sup>0</sup> 10<sup>2</sup> 10<sup>-1</sup> 10<sup>1</sup> 10<sup>3</sup>  $\Delta t$  (s)

Figure 3: Minimum temporal resolutions needed for measuring ion concentrations at the lowest LOD of the APi-TOF-MS system, corresponding to the ion concentrations indicated by the arrows in Fig. 2.

# 3.2 Transmission

267

268

269270

271

272

273

274275

276

277

278279

280

282

283

284

285

Figure 4: Mass-dependent transmission of the APi-TOF-MS system at a high-mass and a low-mass settings (see Table 1), using ion concentrations of approximately 0.1 ppq and  $T_a=0.01~\rm s.$ 

Figure 4 shows the transmission of the APi-TOF-MS system using the high-mass and low-mass setting at  $T_a$ = 0.01 s (see Table 1). For the low-mass setting, a maximum transmission of 1% was achieved at around 250 m/z for positively charged ions (see Table 2). Similar values are observed for ions of ca. 600 m/z for the high-mass setting. These results are similar to those reported for other APi-TOF-MS systems (Heinritzi et al., 2016; Junninen et al., 2010; Leiminger et al., 2019). Negatively charged ions measured using the high-mass setting showed a similar trend as positively charged ions (see Fig. 4). We should note here that measurements at negative polarity were limited to only three species (Br<sup>-</sup>, I<sup>-</sup> and I<sub>2</sub>K<sup>-</sup>), because the mobility-mass calibration standards strongly fragmented at the negative ESI polarity, making them unsuitable for the study of transmission. It should also be noted that the transmission curve may differ for other accumulation times.

The difference in the transmission distribution corresponding to the two conditions is primarily attributed to variations of  $V_{RF,2}$  inside the ion trap. A higher  $V_{RF,2}$  stabilizes the motion of ions with greater m/z, resulting in a transmission curve that shifts towards higher masses under the high-mass setting. The delay of the OA pulses with respect to the ion release from the ion trap was adjusted in the high-mass setting, allowing heavier ions sufficient time to reach the OA (see Fig. S-1).

#### 3.3 Mass resolution

Figure 5 provides measurements of the resolution of the APi-TOF-MS system when feeding it with ions produced by electrospraying a methanol–ammonium sulphate solution (see Fig. S-4). The resolution increases with mass, because heavier ions exhibit longer flight times, which enhances their separation. The lowest detectable mass was identified at 28 m/z, corresponding to  $C_2H_4$ . The highest ion mass that we could produce by electrospray is at 1,244 m/z, corresponding to  $(NH_3)_{35}(H_2SO_4)_{32}(H^+)_3$ . Considering that this molecule carries 3 positive charges, its molecular mass is 3,732 Da. The resolution of the APi-TOF-MS system at the lowest and highest masses used in our measurements were 7,000 and 23,000, respectively.

Figure 5: Mass resolution for different m/z determined by electrospraying a methanol-ammonium sulphate solution.

#### 3.4 Collision-induced dissociation

Figure 6 shows the fraction of the signal produced by the TPAI and THAB dimers over that corresponding to the monomers and their fragments, for varying voltages applied between the ion funnel and the ion trap. The results demonstrate that CID can be controlled by tuning the voltage between those stages. Moreover, the TPAI dimer

dissociates into its monomers at lower voltages than THAB, indicating that the latter is more stable and more resistant to CID.

This is consistent with Density Functional Theory calculations showing that the THAB dimer requires 9 to 11 kcal/mol more energy to dissociate through different pathways with respect to the TPAI dimer (see Supplementary Information for details). Quantum Theory of Atoms in Molecules analyses also show that both dimers are held together by hydrogen bonds formed between the hydrogen atoms of the monomers and the halogen atom (see Supplementary Information for details). These hydrogen-halogen interactions are significantly stronger in the THAB (H-Br bond, 1.55-2.15 kcal/mol) than in the TPAI (H-I bond, 0.52-1.71 kcal/mol) dimer, attributing greater stability and thus higher resistance to dissociation to the former compare to the latter.

Simulations by Zapadinsky et al. (2019) suggest that optimal pressures for CID in APi-TOF-MS systems lie between 10<sup>-2</sup> and 1 mbar; at pressures above 1 mbar, ion-molecule collisions occur too frequently, resulting in drag forces that significantly reduce ion kinetic energy, lowering ion-molecule collision energies to values below the dissociation threshold. On the other hand, at pressures below 10<sup>-2</sup> mbar the frequency of ion-molecule collisions is insufficient to obtain fragments at measurable concentrations. Thus, pressures between the ion funnel (1 mbar) and the ion trap (10<sup>-3</sup> mbar) provide an ideal environment for CID. By adjusting the potential between those two stages, it is possible to manipulate the ion kinetic energies to achieve a sufficient number of collisions with energies exceeding the dissociation threshold.

Figure 6: Fractions of the TPAI (a) and THAB (b) dimers, monomers and other fragmented species as a function of applied voltage between the ion funnel and the ion trap. The fraction represents the relative abundance of each species compared to the total ion population, which is proportional to their collision-induced dissociation efficiency. The dimer concentrations for both ions introduced to the APi-TOF-MS is 0.1 ppq.

# 4. Conclusion

We have characterized a novel APi-TOF-MS system and demonstrated its capability to measure ion concentrations down to the range of 1 part per quintillion ( $10^{-3}$  ppq, i.e., parts per quadrillion; or ~ 30 ions/cm<sup>3</sup> at standard conditions) with a temporal resolution of the order of 1 second. Interestingly, the detection sensitivity can get to 10 parts per sextillion ( $10^{-5}$  ppq; ~0.3 ions/cm<sup>3</sup>) when the temporal resolutions is set to 1 minute, which

is a significant improvement compared existing APi-TOF-MS systems. The high sensitivity of our system is achieved by accumulating ions within an octapole ion trap upstream the TOF chamber, resulting in amplification of the signal without an increase in noise level. This approach is advantageous compared to the conventional approach used currently in APi-TOF-MS systems, whereby signal amplification is achieved by the accumulation of multiple mass spectra that inherently increases both signal and noise simultaneously.

Furthermore, we show that the transmission of the APi-TOF-MS system is up to 1%, which is comparable to other similar mass spectrometers available on the market, and can be optimized for different masses depending on the operating conditions used. We also demonstrate that our system has a mass resolution from 7,000 to 23,000 FWHM for ionic masses ranging from 30 to 1,200 Da, respectively. Furthermore, collision-induced dissociation of sampled ions could be finely controlled by adjusting the potential difference between the funnel and ion trap within the APi.

A clear advantage of the APi-TOF-MS system we investigate here is its versatility, manifested by different operating settings that can be adapted to the needs. For example, when coupled with a chemical ionization source, a possible space-charge build-up from excessive concentrations of reagent ions (e.g., NO<sub>3</sub><sup>-</sup>; 62 Da) inside the ion trap can be mitigated by choosing appropriate RF settings to induce a low-mass cutoff. In addition, the possibility of adding a DMA at the inlet of the APi-TOF-MS system can allow mobility-resolved measurements of low-concentration compounds of interest while preventing space-charge build-up in the ion trap from highly concentrated species that are not relevant. Moreover, the capability for controlled collision-induced dissociation can be utilized for probing the bonding strength of those reagent ions with different types of species.

## Data availability

Schmidt-Ott, F. (2025). Characterization of a High Detection-Sensitivity Atmospheric Pressure Interface Time-of-Flight Mass Spectrometer [data set]. https://doi.org/10.5281/zenodo.15854115

## **Author contribution**

Fabian Schmidt-Ott designed and carried out the measurements and completed most of the manuscript under the guidance of Anne Maisser and George Biskos. Somnath Bhowmick carried out the Density Functional Theory calculations and the Quantum Theory of Atoms in Molecules analyses. Anne Maisser assisted in the design and measurements and provided supervisory input. Alexandros Lekkas and Dimitris Papanastasiou designed and built the APi-TOF-MS and helped to operate it for this study. George Biskos conceptualized the study, supervised the work, and provided input during the writing up of the manuscript. All authors reviewed the manuscript.

## **Competing Interests**

The authors declare that they have no conflict of interest.

# Acknowledgement

This work was supported by the NANO2LAB project that is co-funded by the European Regional Development Fund and the Republic of Cyprus through the Research Innovation Foundation (Strategic Infrastructure), as well as the ASPASIA project as part of the Research and Innovation Foundation (RIF). This work has also received funding from the from the EMME-CARE project, which is financed by the European Union's Horizon 2020 research and innovation program and the Cyprus Government.

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
