# Peer review of "Characterization of a High Detection-Sensitivity Atmospheric"

_EGUsphere, 2025_

## Author Response (AR1)

**Referee #1**

1. Page 1, lines 20–21: I recommend changing the unit from ppq to molecules cm-3, or showing both, when possible. A major motivation for achieving such low LODs is to probe nucleation events (as authors mentioned), where reporting in molecules cm-3 (e.g., 2.5×104 molecules cm-3) is more intuitive and widely used than 1 ppq. That said, using ppq occasionally is fine for readers from the gas-phase or trace gas community. I particularly appreciated Figure 2 and line 237, where both units are shown, this is a good practice and should be applied more consistently.

**Reply**

We agree that molecules or ions cm-3 is a more intuitive unit and that it should be used more consistently throughout the paper. We have made the necessary changes throughout the manuscript, often providing both units for the better understanding.

- 2. **Scientific notation and formatting:** Please check the formatting of numbers throughout the manuscript:
  - o Page 1, line 41: change "10.000" to "10,000" (use a comma for thousands, not a dot)
  - o Page 2, line 57: change "10-5%" to correct scientific notation, e.g., "10-7"

**Reply**

We appreciate the comment of the reviewer here. We have changed the formatting to scientific notation.

- 3. **Define abbreviations at first use.** Several abbreviations are used without definition:
  - o Page 2, line 64: Define "Ts" here rather than in line 65.
  - o Page 2, line 66: Clarify what "ioniAPi-TOF-MS" means, is this a typo?
  - Page 2, line 98: Define "DC" at first mention.
  - o Page 4, line 147: Define "THAB".
  - o Page 5, line 159: Replace the first "TPAI" with "TMAI", possibly a typo.

**Reply**

- The ioniAPi-TOF-MS is the APi-TOF-MS produced by Ionicon Analytik GmbH. We have now clarified this my mentioning the manufacturer in brackets
- Other proposed changes were made.
  - 4. Page 3, line 108: Remove "Fig. 1" after "Gate 1", it is redundant.

**Reply**

The correction has been made.

5. **Page 5, lines 165–166:** Please rephrase this sentence. The LOD is not typically defined as the signal from a single ion striking the detector. Rather, it is defined as the lowest signal that can be reliably distinguished from noise. Your explanation in Supplementary Section S3 is much clearer and more accurate: "The lowest possible signal that the TOF detector of the APi-TOF-MS can measure is that of a single ion striking the detector (Saon)."

**Reply**

We agree with the reviewer here. The correction has been made.

6. Figure 4 and m/z dependence: Figure 4 presents ion transmission for a specific accumulation time (Ta). The authors mention extracting ions at three consecutive time points per spectrum to minimize mass bias during accumulation. I wonder whether the trapping efficiency also varies with m/z, for example, low-m/z ions are unstable due to higher kinetic energy or mobility. Could this be dependent on Ta as well? Comparing transmission curves for different Ta could be insightful and would strengthen the discussion on mass bias.

**Reply**

The reviewer correctly points out that the trapping efficiency over time in the octapole ion trap potentially varies with m/z. We have not investigated this effect in the current work, but we plan to do so in the near future. A sentence highlighting this point has been added in the updated version of the manuscript.

7. **Page 9, line 312:** The authors reference results from Density Functional Theory (DFT) calculations, but note that they are not shown. If possible, please provide a reference to these calculations or supplementary material for readers who may wish to explore them further.

**Reply**

We thank the reviewer for this point. We have added a new section in the supplement (Section S.5)\_ providing a description of the simulations performed to test the stability of THAB and TPAI. Please note that adding this analysis requires the addition of a new co-author, Dr. Somnath Bhowmick.

**Referee #2**

1. Line 66: typo? What do you mean by ioniAPi-TOF-MS

**Reply**

The ioniAPi-TOF-MS is the APi-TOF-MS produced by Ionicon Analytik GmbH. We have now clarified this my mentioning the manufacturer in brackets.

2. You should ensure that each abbreviation is defined before it appears for the first time. For example, in line 87, "ESI" is introduced before its meaning is explained, which you explain them in the following section. In Figure 6, the abbreviations "TPAI" and "THAB" are used in the caption, but they are not consistent with those shown in the figure itself.

**Reply**

We thank the reviewer for pointing this out. We have made necessary changes throughout the entire manuscript.

3. In Table 1, some units are presented alongside the input parameters, while others are placed with the values. Please make the unit placement consistent throughout the table.

**Reply**

We have made necessary corrections, improving the consistency on how we refer to the units through the manuscript.

4. In general, did you describe what kind of solutions that you made for ESI to generate the ions shown in Table 2? I did not find them. I think it is better to include this information in paper.

**Reply**

We agree with the suggestion of the referee to add the details on the ESI solutions. We have added a relevant part in section 2.2.

5. In line 147, line 256: How did you operate DMS to select single ions with different mass to charge ratio? You only mentioned the parameters to select the THAB monomer, how about others that you listed in Table 1.

**Reply**

We understand that the reviewer here wanted to write DMA instead of DMS, and that parameters for the THAB monomer are the ones provided in Table 2 and not Table 1.

The parameters to select monomers and dimers other than THAB was only the potential difference between the two plates of the DMA electrodes. We have modified the text accordingly for clarification.

6. Line 198 and 200: Water molecules can also have an effect on the binding energies of ion-molecular clusters, which can help to stabilize the ion-molecular clusters. Please have a look at the paper 'Computational Comparison of Different Reagent Ions in the Chemical Ionization of Oxidized Multifunctional Compounds'.

**Reply**

The reviewer brings up an interesting point, that ion-molecular clusters can be stabilized by binding to water molecules. Since water is presumably absent in our measurements (the electrospray solutions we prepared in acetonitrile), we expect that the effect of water on the binding is negligible in this study. However, for ambient measurement, indeed the stabilization by water molecules can play an important role, where fragmentation of these water-bound clusters is possibly reached at far lower energies than seen in this study. We have included the reference and pointed out, that the presence of water can have a significant effect on binding energies.

7. In section 3.2. How did you operate the system to define the transmission? What kind of ions with what mass that you used to define the transmission? Do you mean all the ions that you listed in table 2?

This paragraph has now been updated, and addresses the point of the reviewer here clarifying the setting of the API-TOF-MS during the transmission measurements.

8. Line 315: Is there any reference here?

**Reply**

Given that referee #1 made a similar comment, we have added chapter S.5 in the supplement providing an in-depth explanation on the simulations performed to test the stability of THAB and TPAI to the supplement.

9. Line 315-319: What would you obtain in mass spectrum if the dimer ions dissociate? If I understand correctly, I would say, it should be one neutral molecule and one ion-molecule cluster. That means it should not be the neutral monomer in your figure 6. For example, the possible combination should be (C12H28N)2I-, (C12H28N)I- and (C7H16N)I- in panel a, but not (C12H28N)2I-, C12H28N and C7H16N. The same case for panel b.

**Reply**

We have specified the chemical formulas of possible products of the dissociation of the THAB and TPAI dimers in Figure S-8 in the supplement.

Based on comment 2 of the reviewer, we have chosen to change the terminology in the legend to "TPAI" and "THAB" monomer/dimer, making the chemical formulas redundant in Fig. 6.

---

## Author Response (AR2)

**Supplementary information**

**Characterization of a High Detection-Sensitivity Atmospheric Pressure Interface Time-of-Flight Mass Spectrometer**

Fabian Schmidt-Ott1,2, Somnath Bhowmick1, Alexandros Lekkas,3 Dimitris Papanastasiou3, Anne Maisser1, George Biskos1,4

*Corresponding authors*: Fabian Schmidt-Ott (f.schmidt-ott@cyi.ac.cy) or George Biskos (g.biskos@cyi.ac.cy or g.biskos@tudelft.nl)

**S.1 Timing for accelerating ions into the TOF**

Once released from the ion trap, ions travel approximately 0.5 m before reaching the orthogonal acceleration (OA) stage, where ion packets are injected into the TOF chamber in synchronization with their release. Because ions require a finite time to reach the OA stage, a delay occurs between their release from the trap and their orthogonal acceleration.

Furthermore, the delay is mass-dependent: larger ions take longer to travel from the ion trap exit to the OA stage and thus experience greater delays than smaller ions. This mass dependency in delay arises from the same principle that governs ion separation in the TOF chamber - larger ions experience lower acceleration across an electric field and therefore travel at lower velocities. Accurate synchronization of ion release from the trap with the OA pulsing is therefore essential to prevent ion losses due to mismatched timing.

To characterize this mass-dependent delay, transmission was measured using ion samples with distinctly different masses (see Fig. S-1). An ion of 74 Da showed optimal transmission with a 40  $\mu$ s delay, whereas an ion of 690 Da required a 110  $\mu$ s delay. Based on these results, the OA delays summarized in Table 1 were selected, with shorter delays applied to the low-mass range and longer delays to the high-mass range.

**Figure S-1.** Normalized transmission as a function of OA timing. The transmission was normalized with respect to the highest value, because the transmission was different for each ion standard.

<sup>1 Climate and Atmosphere Research Centre, The Cyprus Institute, 2121 Nicosia, Cyprus

<sup>2 Institute for Atmospheric and Earth System Research, University of Helsinki, 00014, Helsinki, Finland

<sup>3 Fasmatech Science and Technology SA, TESPA Lefkippos, NCSR Demokritos, 15310 Athens, Greece

<sup>4 Faculty of Civil Engineering and Geosciences, Delft University of Technology, 2628 CN, Delft, The Netherlands

**S.2 Ion production stability**

Figure S-2 shows the temporal variation of the THAB concentration produced by electrospray. Since all measurements required a stable source of ions, we ensured that the relative standard deviation was approximately 10% or less in all cases.

**Figure S-2.** Stability of THAB monomer production when electrosprayed at different concentrations.

**S.3 Single-ion measurements**

The lowest possible signal that the TOF detector (i.e., a MagneTOF® detector) of the APi-TOF MS can measure is that of a single ion striking the detector ( $S_{ion}$ ; Simke et al., 2024). As indicated by the recorded spectrum shown in Figure S-3, this signal is 60 a.u. and lies well above the noise level of the detector that has a standard deviation ( $\sigma_{noise}$ ) of 2 a.u. Consequently, the SNR (which is equal to  $S_{ion}/\sigma_{noise}$ ) corresponding to the single-ion peak is approximately 30, which is line with the recommended minimum threshold of 10 (Gross, 2006). We should note here that the value of  $S_{ion}$  is used to determine the count rate and from that the transmission of the APi-TOF-MS system by Eq. 1 in the main manuscript.

**Figure S-3.** A single non-averaged mass spectrum showing the signal induced by a single THAB monomer striking the TOF detector.

**S.4 Sulphuric acid – amine clusters**

Figure S-4 shows the mass spectrum of ionic clusters comprised of amine (NH3) and sulphuric acid (H2SO4), with the general formulation (NH3)x(H2SO4)x-y(H+)y. These clusters were produced by electrospraying a solution of ammonium sulphate in methanol as described by Waller et al. (2019).

**Figure S-4.** Amine – sulphuric acid clusters generated by electrospraying an ammonium sulphate methanol solution. Key: A stands for  $NH_3$ . and B for  $H_2SO_4$ .

Figure S-5 shows the CID as the TPAI monomer brakes into its fragments at varying voltages across the funnel and ion trap, corresponding to different kinetic energies. For this measurement, only the TPAI monomer was selected by the DMA upstream the APi-TOF-MS system. Fragmentation onset here is at 25 V, which is lower than the onset observed for the TPAI monomer when the dimer is present, which occurs at 45 V, as shown in Figure 6 in the main manuscript. Apparently, the dimer breaks first, before the molecule itself can fragment in smaller pieces. This is consistent with the fact that ionic bonds holding the dimer together are generally weaker than covalent bondings within the monomer molecules.

**Figure S-5.** Collision-induced dissociation of TPAI monomers at different voltages across the funnel and ion trap, shown as the fraction of the species with respect to the total ion signal. The monomer concentration introduced to the API-TOF MS is  $5 \times 10^4$  ions/cm3.

**S.5 TPAI and THAB dimer stability**

We used the ABCluster programme in combination with the Gaussian 16 *ab initio* electronic structure calculation package to determine ground state equilibrium geometries of THAB and TPAI monomers and dimers (Frisch et al., 2019; Zhang & Dolg, 2015). ABCluster employs the artificial bee colony algorithm to efficiently explore the potential energy surface of a molecular system. The structures obtained from ABCluster were then fully optimized using density functional theory (DFT) with the ωB97X-D [7] hybrid functional and Def2-TZVP basis set [8] to identify ground state equilibrium geometries of THAB and TPAI monomers and dimers (Chai & Head-Gordon, 2008; Jones, 2015; Weigend & Ahlrichs, 2005). Similar computational approaches have been shown to reproduce molecular geometries and electronic properties in close agreement with the experimental results for similar systems (Domingos et al., 2021).

**Figure S-6.** Optimized equilibrium geometry of the THAB dimer calculated at the  $\omega B97X$ -D/def2-TZVP level of theory. Atoms participating in the hydrogen bond (Br and H) are labeled, with partial atomic charges indicated in brackets. The hydrogen bond is depicted as a dashed line, and the corresponding interatomic distance (in Å) is shown in green.

**Figure S-7.** Optimized equilibrium geometry of the TPAI dimer calculated at the  $\omega$ B97X-D/def2-TZVP level of theory. Atoms participating in the hydrogen bond (I and H) are labeled, with partial atomic charges indicated in brackets. The hydrogen bond is depicted as a dashed line, and the corresponding interatomic distance (in Å) is shown in green.

The optimized geometries of THAB and TPAI dimers are illustrated in Figures S1 and S2. In both systems, the most dominating stabilizing interactions between the two monomer units are hydrogen bonds between the halogen atom (Br or I) and the hydrogen atoms of the monomers. The presence and nature of these hydrogen bonds were confirmed by Quantum Theory of Atoms in Molecules analysis performed using the Multiwfn program (Lu & Chen, 2012; Richard & Bader, 1990). Partial atomic charges on the atoms involved in these hydrogen bonds were obtained using the Natural Population Analysis method, and the values for the H and halogen atoms were found to be similar in both dimers (Reed et al., 1985).

Structurally, the TPAI dimer contains eight hydrogen bonds compared to six in the THAB dimer. However, the average hydrogen bond length in the TPAI dimer is longer. Specifically, in the TPAI dimer, two hydrogen bonds are very long (3.52 Å) and four others fall in the range of 2.84-2.91 Å. In contrast, in the THAB dimer, four hydrogen bonds are significantly shorter ( $\approx$ 2.65 Å), and the remaining two are slightly longer ( $\approx$ 2.89 Å). This trend suggests that the hydrogen bonds in TPAI dimer are generally weaker than those in the THAB dimer. The above statement is supported by the quantitative estimation of hydrogen bond strength obtained from Espinosa's relation (Espinosa et al., 1998). For the THAB dimer, the hydrogen bond energies range from 1.55 to 2.15 kcal/mol, whereas for the TPAI dimer, they range from 0.52 to 1.71 kcal/mol. When expressed as the average hydrogen bond energy per bond, the THAB dimer exhibits a significantly higher value (1.94 kcal/mol) compared to TPAI dimer (1.13 kcal/mol).

An alternative approach to evaluate the relative stability of the THAB and TPAI dimers is to compare their total energies with those of their dissociated products. In APi-TOF-MS, collision-induced dissociation of both dimers yields the product sets shown in reaction paths 1 and 2 shown in Figure S-8 below.

$$(C_{28}H_{60}N)_2Br^+ \begin{cases} (C_{28}H_{60}N)^+ + (C_{28}H_{60}N)Br & (1a) & \Delta E = 40.04 \ kcal/mol \\ \\ 2(C_{28}H_{60}N)^+ + Br^- & (1b) & \Delta E = 122.20 \ kcal/mol \\ \\ (C_{12}H_{28}N)_2I^+ \end{cases}$$

$$(C_{12}H_{28}N)^+ + (C_{12}H_{28}N)I \qquad (2a) \qquad \Delta E = 31.14 \ kcal/mol \\ \\ (C_{12}H_{28}N)^+ + I^- \qquad (2b) \qquad \Delta E = 111.45 \ kcal/mol \\ \end{cases}$$

 $\Delta E$  = sum of energies of the dissociated products – energy of the dimer

**Figure S-8.** Dissociation channels of THAB and TPAI dimers with associated stabilization energy values ( $\Delta E$ ).

If the total energy of the intact dimer (left-hand side of reaction paths 1 and 2) is lower than the sum of the energies of its dissociation products (right-hand side of the reaction paths), the dimer is considered stable with respect to dissociation. The stabilization energy,  $\Delta E$ , reported in Figure S-8, is the energy difference between the sum of energies of the products and intact dimer, and therefore, provides a direct measure of this stability; i.e., larger and positive  $\Delta E$  values correspond to greater overall stability.

DFT calculations show that  $\Delta E$  is positive for all dissociation channels of both dimers, consistent with their behavior in our APi-TOF-MS system. For a particular dissociation channel, the  $\Delta E$  values for the THAB dimer are 9 to 11 kcal/mol larger than those for the TPAI dimer, indicating greater stability. The enhanced stability allows the THAB dimer to resist higher collisional energies before dissociating, whereas the weaker hydrogen bonding and lower  $\Delta E$  values in the TPAI dimer make it more prone to fragmentation. As a result, TPAI dimers dissociate at lower collision energies in APi-TOF-MS measurements, which is in agreement with our experimental observation.

**References**

- Chai, J.-D., & Head-Gordon, M. (2008). Long-range corrected hybrid density functionals with damped atom—atom dispersion corrections. *Physical Chemistry Chemical Physics*, 10(44), 6615. https://doi.org/10.1039/b810189b
- Domingos, S. R., Pérez, C., Kreienborg, N. M., Merten, C., & Schnell, M. (2021). Dynamic chiral self-recognition in aromatic dimers of styrene oxide revealed by rotational spectroscopy. *Communications Chemistry*, *4*(1), 32. https://doi.org/10.1038/s42004-021-00468-4
- Espinosa, E., Molins, E., & Lecomte, C. (1998). Hydrogen bond strengths revealed by topological analyses of experimentally observed electron densities. *Chemical Physics Letters*, 285(3–4), 170–173. https://doi.org/10.1016/S0009-2614(98)00036-0
- Gaussian 16, Revision C.01, Frisch, M. J.; Trucks, G. W.; Schlegel, H. B.; Scuseria, G. E.; Robb, M. A.; Cheeseman, J. R.; Scalmani, G.; Barone, V.; Petersson, G. A.; Nakatsuji, H.; Li, X.; Caricato, M.; Marenich, A. V.; Bloino, J.; Janesko, B. G.; Gomperts, R.; Mennucci, B.; Hratchian, H. P.; Ortiz, J. V.; Izmaylov, A. F.; Sonnenberg, J. L.; Williams-Young, D.; Ding, F.; Lipparini, F.; Egidi, F.; Goings, J.; Peng, B.; Petrone, A.; Henderson, T.; Ranasinghe, D.; Zakrzewski, V. G.; Gao, J.; Rega, N.; Zheng, G.; Liang, W.; Hada, M.; Ehara, M.; Toyota, K.; Fukuda, R.; Hasegawa, J.; Ishida, M.; Nakajima, T.; Honda, Y.; Kitao, O.; Nakai, H.; Vreven, T.; Throssell, K.; Montgomery, J. A., Jr.; Peralta, J. E.; Ogliaro, F.; Bearpark, M. J.; Heyd, J. J.; Brothers, E. N.; Kudin, K. N.; Staroverov, V. N.; Keith, T. A.; Kobayashi, R.; Normand, J.; Raghavachari, K.; Rendell, A. P.; Burant, J. C.; Iyengar, S. S.; Tomasi, J.; Cossi, M.; Millam, J. M.; Klene, M.; Adamo, C.; Cammi, R.; Ochterski, J. W.; Martin, R. L.;

- Morokuma, K.; Farkas, O.; Foresman, J. B.; Fox, D. J. (2016) Gaussian, Inc., Wallingford CT.Gross, J. H. (2006). *Mass spectrometry: A textbook*. Springer Science & Business Media.
- Jones, R. O. (2015). Density functional theory: Its origins, rise to prominence, and future. *Reviews of Modern Physics*, 87(3), 897–923. https://doi.org/10.1103/RevModPhys.87.897
- Lu, T., & Chen, F. (2012). Multiwfn: A multifunctional wavefunction analyzer. *Journal of Computational Chemistry*, *33*(5), 580–592. https://doi.org/10.1002/jcc.22885
- Reed, A. E., Weinstock, R. B., & Weinhold, F. (1985). Natural population analysis. *The Journal of Chemical Physics*, 83(2), 735–746. https://doi.org/10.1063/1.449486
- Richard, F., & Bader, R. (1990). Atoms in molecules: A quantum theory.
- Simke, F., Fischer, P., & Schweikhard, L. (2024). Evaluation of MagneTOF detector signals for the determination of many-ion bunches. *International Journal of Mass Spectrometry*, *506*, 117337. https://doi.org/10.1016/j.ijms.2024.117337
- Waller, S. E., Yang, Y., Castracane, E., Kreinbihl, J. J., Nickson, K. A., & Johnson, C. J. (2019). Electrospray Ionization–Based Synthesis and Validation of Amine-Sulfuric Acid Clusters of Relevance to Atmospheric New Particle Formation. *Journal of the American Society for Mass Spectrometry*, 30(11), 2267–2277. https://doi.org/10.1007/s13361-019-02322-3
- Weigend, F., & Ahlrichs, R. (2005). Balanced basis sets of split valence, triple zeta valence and quadruple zeta valence quality for H to Rn: Design and assessment of accuracy. *Physical Chemistry Chemical Physics*, 7(18), 3297. https://doi.org/10.1039/b508541a
- Zhang, J., & Dolg, M. (2015). ABCluster: The artificial bee colony algorithm for cluster global optimization. *Physical Chemistry Chemical Physics*, 17(37), 24173–24181. https://doi.org/10.1039/C5CP04060D